# Dust-Acoustic Rogue Waves in an Electron-Positron-Ion-Dust Plasma Medium

**Md. Habibur Rahman** [1,*], **Nure Alam Chowdhury** [2], **Abdul Mannan** [1] **and A. A. Mamun** [1]

1 Department of Physics, Jahangirnagar University, Dhaka 1342, Bangladesh; abdulmannan@juniv.edu (A.M.); mamun_phys@juniv.edu (A.A.M.)
2 Plasma Physics Division, Atomic Energy Centre, Dhaka 1000, Bangladesh; nurealam1743phy@gmail.com
* Correspondence: rahman1992phy@gmail.com

**Abstract:** In this work, the modulational instability of dust-acoustic (DA) waves (DAWs) is theoretically studied in a four-component plasma medium with electrons, positrons, ions, and negative dust grains. The nonlinear and dispersive coefficients of the nonlinear Schrödinger equation (NLSE) are used to recognize the stable and unstable parametric regimes of the DAWs. It can be seen from the numerical analysis that the amplitude of the DA rogue waves decreases with increasing populations of positrons and ions. It is also observed that the direction of the variation of the critical wave number is independent (dependent) of the sign (magnitude) of $q$. The applications of the outcomes from the present investigation are briefly addressed.

**Keywords:** NLSE; reductive perturbation method; rogue waves

---

## 1. Introduction

Electron–positron–ion-dust (EPID) plasma has been identified in the galactic centre [1], Saturn's magnetosphere [2,3], Jupiter's magnetosphere [3,4], the pulsar magneto-sphere [5–8], supernova environments [8,9], interstellar medium [10–14], cometary tails [15–17], the solar atmosphere [17–20], and laboratory experiments [21–25]. There are many electron–ion plasma systems in which positron or charged dust species or both occur naturally due to many mechanisms (viz., pair production [18], thermal heating [26,27], and radiative heating [28], etc.). The dynamics of the EPID plasma medium (EPIDPM) and associated electrostatic nonlinear waves have rigorously changed due to the existence of the light positron and heavy dust grains in the EPIDPM [29–32]. The signature of the positron in the EPIDPM has encouraged many authors to examine the nonlinear electrostatic pulses in the EPIDPM [29,30]. Banerjee and Maitra [29] considered a four-component EPIDPM, studied the electrostatic potential structures in the presence of massive dust grains and light positrons, and observed that the height of the potential structures increases with increasing dust number density but decreases with increasing positron number density. Paul and Bandyopadhyay [30] demonstrated dust–ion-acoustic waves in an EPIDPM and showed that only positive super-solitons can exist.

Highly energetic particles have been observed in Saturn's magnetosphere [2,3], Jupiter's magnetosphere [3,4], the vicinity of the Moon [33], Earth's bow-shock [34], and galaxy clusters [35], etc. Renyi [36] first noticed the deviation of these particles from a Maxwellian–Boltzman distribution, and finally Tsallis [37] generalized the non-extensive $q$-distribution to explain these particles. The parameter $q$ in the $q$-distribution describes the deviation of these particles from a Maxwellian–Boltzman distribution, and when $q \rightarrow 1$, Tsallis distribution coincides with the Maxwell–Boltzmann distribution [38–40]. Eslami et al. [38] examined the dust-acoustic (DA) solitary waves (DA-SWs) in the presence of non-extensive plasma species. Roy et al. [39] studied the DA shock waves (DA-SHWs) in a three-component dusty plasma featuring non-extensive electrons and observed that the height of the DA-SHWs decreases with $q$.

---

The modulational instability (MI) of electrostatic waves and associated rogue waves are governed by the nonlinear Schrödinger equation (NLSE) [41–43]. Bains et al. [41] studied the MI of the DA waves (DAWs) by deriving the NLSE in a three-component dusty plasma with non-extensive plasma species and found that the critical wave number ($k_c$), which determines the stability of the electrostatic waves, increases with increasing non-extensivity of the plasma species. Moslem et al. [42] analyzed the DA rogue waves (DA-RWs) by considering $q$-distributed plasma species. Rahman et al. [43] examined DA-RWs in a multi-component dusty plasma and observed that the temperature of the ion enhances the height of the DA-RWs.

Recently, Esfandyari-Kalejahi et al. [44] investigated the electrostatic DA-SWs in an EPIDPM and observed that the amplitude of DA-SWs increases with increasing charge of the dust grains. Jehan et al. [45] studied DA-SWs in a four-component dusty plasma by considering inertial massive dust grains and inertialess iso-thermal electrons, positrons, and ions. To the best knowledge of the authors, no theoretical investigation has been made to understand the stability of the DAWs in a four-component EPIDPM. Therefore, in this paper, we study the MIs of DAWs and the formation of DA-RWs in an EPIDPM.

The rest of the paper is organized as follows. The governing equations are presented in Section 2. The MIs of DAWs and associated DA-RWs are given in Section 3. Finally, a brief conclusion is provided in Section 4.

## 2. Governing Equations

We consider the propagation of DAWs in an unmagnetized EPIDPM with inertial, warm, negatively-charged, massive dust grains (mass $m_d$; charge $q_d = -Z_d e$; temperature $T_d$; number density $N_d$) and inertialess $q$-distributed electrons (mass $m_e$; charge $-e$; temperature $T_e$; number density $N_e$), positrons (mass $m_p$; charge $+e$; temperature $T_p$; number density $N_p$), and ions (mass $m_i$; charge $q_i = +Z_i e$; temperature $T_i$; number density $N_i$), where $Z_d$ ($Z_i$) is the number of electrons (protons) residing in negatively (positively) charged dust grains (ions). The dynamics of the EPIDPM is governed by these equations:

$$\frac{\partial N_d}{\partial T} + \frac{\partial (N_d U_d)}{\partial X} = 0, \tag{1}$$

$$\frac{\partial U_d}{\partial T} + U_d \frac{\partial U_d}{\partial X} + \frac{1}{m_d N_d} \frac{\partial P_d}{\partial X} = \frac{Z_d e}{m_d} \frac{\partial \tilde{\varphi}}{\partial X}, \tag{2}$$

$$\frac{\partial^2 \tilde{\varphi}}{\partial X^2} = 4\pi e [N_e - N_p - Z_i N_i + Z_d N_d], \tag{3}$$

where $U_d$ is the dust fluid speed, $P_d$ is the pressure of the dust grains, and $\tilde{\varphi}$ represents the electrostatic wave potential. To obtain the normalized form of Equations (1)–(3), we introduce the normalized parameters; namely, $n_d \rightarrow N_d/n_{d0}$, $n_e \rightarrow N_e/n_{e0}$, $n_p \rightarrow N_p/n_{p0}$, and $n_i \rightarrow N_i/n_{i0}$, where $n_{d0}$, $n_{e0}$, $n_{p0}$, and $n_{i0}$ are the equilibrium number densities of the dust grains, electrons, positrons, and ions, respectively; $u_d \rightarrow U_d/C_d$, where $C_d = (Z_d k_B T_i/m_d)^{1/2}$ and $k_B$ is the Boltzmann constant; $\varphi \rightarrow \tilde{\varphi} e/k_B T_i$; $t = T/\omega_{pd}^{-1}$, where $\omega_{pd}^{-1} = (m_d/4\pi Z_d^2 e^2 n_{d0})^{1/2}$; and $x = X/\lambda_{Dd}$, where $\lambda_{Dd} = (k_B T_i/4\pi Z_d n_{d0} e^2)^{1/2}$. The pressure term of the dust grains can be recognized as $P_d = P_{d0}(N_d/n_{d0})^{\gamma}$, with $P_{d0} = n_{d0} k_B T_d$ and $\gamma = (N+2)/N$, where $N$ is the degree of freedom, and for the one-dimensional case, $N = 1$, then $\gamma = 3$. The equilibrium quasi-neutrality condition can be written as $n_{e0} + Z_d n_{d0} \simeq n_{p0} + Z_i n_{i0}$. Now, after employing the normalizing parameters, we can write the normalized form of Equations (1)–(3) as

$$\frac{\partial n_d}{\partial t} + \frac{\partial (n_d u_d)}{\partial x} = 0, \tag{4}$$

$$\frac{\partial u_d}{\partial t} + u_d \frac{\partial u_d}{\partial x} + \delta n_d \frac{\partial n_d}{\partial x} = \frac{\partial \varphi}{\partial x}, \tag{5}$$

$$\frac{\partial^2 \varphi}{\partial x^2} = (\mu_p + \mu_i - 1) n_e - \mu_p n_p - \mu_i n_i + n_d. \tag{6}$$

Other plasma parameters are considered as $\delta = 3T_d/Z_dT_i$, $\mu_p = n_{p0}/Z_dn_{d0}$, and $\mu_i = Z_in_{i0}/Z_dn_{d0}$. Now, the non-extensive $q$-distributed electron, positron, and ion number densities can be expressed as [37,43]

$$n_e = \left[1 + (q_e - 1)\sigma\varphi\right]^{\frac{q_e+1}{2(q_e-1)}},\tag{7}$$

$$n_p = \left[1 - (q_p - 1)\alpha\varphi\right]^{\frac{q_p+1}{2(q_p-1)}},\tag{8}$$

$$n_i = \left[1 - (q_i - 1)\varphi\right]^{\frac{q_i+1}{2(q_i-1)}},\tag{9}$$

where $\sigma = T_i/T_e$ and $\alpha = T_i/T_p$. For simplicity, we have considered $q_e = q_p = q_i = q$, where $q$ is the non-extensive parameter defining the degree of non-extensivity of plasma species; i.e., $q = 1$ corresponds to the Maxwellian distribution, and $q < 1$ ($q > 1$) corresponds to the super-extensivity (sub-extensivity). We note that we are interested in DAWs with a frequency of 10 to 100 Hz [20] and that the charging frequency of the dust species in electron–ion plasma is on the order of $10^6$ [20]. This clearly indicates that the dust charge fluctuation is important only for the waves whose frequency is comparable to the dust charging frequency. Thus, the dust charging time scale is completely negligible in comparison with that of the DAWs, and the effect of the dust charge fluctuation can be reasonably neglected in any kind of study of the DAWs. We further note that the electron species is assumed to follow a non-extensive distribution. Thus, the estimation of characteristic times of electron thermalization for the plasma system under consideration is irrelevant. Now, by expanding Equations (7)–(9) to the third-order in $\varphi$ and thus substituting these expansions into Equation (6), one can easily write

$$\frac{\partial^2\varphi}{\partial x^2} + 1 = n_d + g_1\varphi + g_2\varphi^2 + g_3\varphi^3 + \cdots,\tag{10}$$

where

$$g_1 = [(q+1)\{(\mu_p + \mu_i - 1)\sigma + \mu_p\alpha + \mu_i\}]/2,$$
$$g_2 = [(q+1)(q-3)\{(1 - \mu_p - \mu_i)\sigma^2 + \mu_p\alpha^2 + \mu_i\}]/8,$$
$$g_3 = [(q+1)(q-3)(3q-5)\{(\mu_p + \mu_i - 1)\sigma^3 + \mu_p\alpha^3 + \mu_i\}]/48.$$

By employing the reductive perturbation method, one can easily derive the NLSE [46–48]. For the derivation of the NLSE, the stretched coordinates are considered as $\xi = \epsilon(x - v_gt)$ and $\tau = \epsilon^2 t$, where $\epsilon$ is small parameter (i.e., $\epsilon \ll 1$) and $v_g$ is the group velocity of the DAWs [46–48]. The dependent variables can be represented as [46–48]

$$n_d = 1 + \sum_{m=1}^{\infty} \epsilon^m \sum_{l=-\infty}^{\infty} n_{dl}^{(m)}(\xi, \tau)\exp[il(kx - \omega t)],\tag{11}$$

$$u_d = \sum_{m=1}^{\infty} \epsilon^m \sum_{l=-\infty}^{\infty} u_{dl}^{(m)}(\xi, \tau)\exp[il(kx - \omega t)],\tag{12}$$

$$\varphi = \sum_{m=1}^{\infty} \epsilon^m \sum_{l=-\infty}^{\infty} \varphi_l^{(m)}(\xi, \tau)\exp[il(kx - \omega t)],\tag{13}$$

where $k$ ($\omega$) stands for the carrier wave number (frequency). The derivative operators are [46–48]

$$\frac{\partial}{\partial t} \rightarrow \frac{\partial}{\partial t} - \epsilon v_g\frac{\partial}{\partial \xi} + \epsilon^2\frac{\partial}{\partial \tau}, \quad \frac{\partial}{\partial x} \rightarrow \frac{\partial}{\partial x} + \epsilon\frac{\partial}{\partial \xi}.\tag{14}$$

Now, by substituting Equations (11)–(14) into Equations (4), (5), and (10), and under consideration of $m = 1$ with $l = 1$, we can write Equations (4) and (5) as

$$n_{d1}^{(1)} = \frac{k^2}{\delta k^2 - \omega^2} \varphi_1^{(1)}, \qquad u_{d1}^{(1)} = \frac{k\omega}{\delta k^2 - \omega^2} \varphi_1^{(1)}, \tag{15}$$

and the dispersion relation of DAWs is

$$\omega^2 = \frac{k^2}{g_1 + k^2} + \delta k^2. \tag{16}$$

The second-order ($m = 2$ with $l = 1$) equations are given by

$$n_{d1}^{(2)} = \frac{k^2}{\delta k^2 - \omega^2} \varphi_1^{(2)} - \frac{2i\omega v_g k^2 - 2ik\omega^2}{\delta^2 k^4 - 2\delta k^2 \omega^2 + \omega^4} \frac{\partial \varphi_1^{(1)}}{\partial \xi}, \tag{17}$$

$$u_{d1}^{(2)} = \frac{k\omega}{\delta k^2 - \omega^2} \varphi_1^{(2)} - \frac{iv_g k\omega^2 + i\delta v_g k^2 - i\omega^3 - i\delta \omega k^2}{\delta^2 k^4 - 2\delta k^2 \omega^2 + \omega^4} \frac{\partial \varphi_1^{(1)}}{\partial \xi}, \tag{18}$$

with the compatibility condition

$$v_g = \frac{\omega^2 - \delta^2 k^4 + 2\delta k^2 \omega^2 - \omega^4}{k\omega}. \tag{19}$$

The coefficients of $\epsilon$ under consideration of $m = 2$ and $l = 2$ provide the following relations:

$$n_{d2}^{(2)} = B_1 |\varphi_1^{(1)}|^2, \quad u_{d2}^{(2)} = B_2 |\varphi_1^{(1)}|^2, \quad \varphi_2^{(2)} = B_3 |\varphi_1^{(1)}|^2, \tag{20}$$

where

$$B_1 = \frac{2B_3 \delta^2 k^6 - 4B_3 \delta \omega^2 k^4 + 2B_3 k^2 \omega^4 - 3\omega^2 k^4 - \delta k^6}{2\delta^3 k^6 - 4\delta^2 \omega^2 k^4 + 2\delta k^2 \omega^4 - 2\delta^2 \omega^2 k^4 + 4\delta k^2 \omega^4 - 2\omega^6},$$

$$B_2 = \frac{2B_1 \delta k (\delta^2 k^4 - 2\delta k^2 \omega^2 + \omega^4) + \delta k^5 + \omega^2 k^3 - 2B_3 k (\delta^2 k^4 - 2\delta k^2 \omega^2 + \omega^4)}{2\omega \delta^2 k^4 - 4\delta k^2 \omega^3 + 2\omega^5},$$

$$B_3 = \frac{3\omega^2 k^4 + \delta k^6 - (2g_2 \delta k^2 - 2g_2 \omega^2)(\delta^2 k^4 - 2\delta k^2 \omega^2 + \omega^4)}{6\delta^3 k^8 - 12\delta^2 k^6 \omega^2 + 6\delta k^4 \omega^4 - 6\delta^2 k^2 \omega^6 + 12k^4 \omega^4 - 6k^2 \omega^6}.$$

Now, we consider the expressions for ($m = 3$ with $l = 0$) and ($m = 2$ with $l = 0$), which lead to the zeroth harmonic modes. Thus, we obtain

$$n_{d0}^{(2)} = B_4 |\varphi_1^{(1)}|^2, \quad u_{d0}^{(2)} = B_5 |\varphi_1^{(1)}|^2, \quad \varphi_0^{(2)} = B_6 |\varphi_1^{(1)}|^2, \tag{21}$$

where

$$B_4 = \frac{B_6 \delta^2 k^4 - 2\delta B_6 k^2 \omega^2 + B_6 \omega^4 - 2v_g \omega k^3 - \delta k^4 - k^2 \omega^2}{(\delta^2 k^4 - 2\delta k^2 \omega^2 + \omega^4)(\delta - v_g^2)},$$

$$B_5 = \frac{B_4 \delta^3 k^4 - 2B_4 \delta^2 k^2 \omega^2 + B_4 \delta \omega^4 - B_6 \delta^2 k^4 + 2B_6 \delta k^2 \omega^2 - B_6 \omega^4 + \delta k^4 + k^2 \omega^2}{v_g \delta^2 k^4 - 2v_g \delta k^2 \omega^2 + v_g \omega^4},$$

$$B_6 = \frac{2v_g \omega k^3 + \delta k^4 + k^2 \omega^2 - (2g_2 \delta^2 k^4 - 4g_2 \delta k^2 \omega^2 + 2g_2 \omega^4)(\delta - v_g^2)}{(\delta^2 k^4 - 2\delta k^2 \omega^2 + \omega^4)(1 + g_1 \delta - g_1 v_g^2)}.$$

Finally, the coefficients of $\epsilon$ under consideration of $m = 3$ and $l = 1$ and with the help of (15)–(21) provide the NLSE [46–48]:

$$i\frac{\partial \phi}{\partial \tau} + P\frac{\partial^2 \phi}{\partial \xi^2} + Q|\phi|^2 \phi = 0, \tag{22}$$

where $\phi = \varphi_1^{(1)}$ for simplicity. In Equation (22), the dispersion coefficient ($P$) can be written as

$$P = \frac{(v_g k - \omega)(\omega^3 - 3v_g k\omega^2 + 3\delta\omega k^2 - v_g\delta k^3) - (\delta k^2 - \omega^2)(\delta^2 k^4 - 2\delta k^2\omega^2 + \omega^4)}{2\omega\delta k^4 - 2k^2\omega^3},$$

and the nonlinear coefficient ($Q$) can be written as

$$Q = \frac{3g_3(\delta^2 k^4 - 2\delta k^2\omega^2 + \omega^4) + 2g_2(\delta^2 k^4 - 2\delta k^2\omega^2 + \omega^4)(B_3 + B_6) - F}{2\omega k^2},$$

where $F = 2\omega B_2 k^3 + 2\omega B_5 k^3 - \delta B_1 k^4 - B_1 k^2\omega^2 - \delta B_4 k^4 - B_4 k^2\omega^2$.

## 3. Modulational Instability and Rogue Waves

When $P$ and $Q$ of the NLSE (22) have the same sign (i.e., $P/Q > 0$) then the DAWs are modulationally unstable, and when the $P$ and $Q$ values of the NLSE (22) have the opposite sign (i.e., $P/Q < 0$), then the DAWs are modulationally stable [46–49]. The point at which transition of the $P/Q$ curve intersects with the $k$-axis is known as the critical wave number $k$ (=$k_c$). It may be noted here that for typical dusty plasma, a number of authors have considered $N_{i0} \simeq (10^7–10^{13})$ cm$^{-3}$ [19–21,50–53], $N_{e0} \simeq (10^7–10^{13})$ cm$^{-3}$ [18–21,50–53], $N_{p0} \simeq 7 \times 10^7$ cm$^{-3}$ [18,53], $k_B T_e \simeq (3–8)$ eV [19–21,50–53], $k_B T_p \simeq 8$ eV [18,53], $k_B T_i \simeq (0.2–1)$ eV [19–21,50–53], $k_B T_d \simeq (0.1–0.3)$ eV [19–21,50–53], and $Z_d \simeq 10^3–10^4$ [19–21,50–53], and for laboratory edge dusty plasma, $N_{e0} \simeq N_{i0} \simeq 10^{13}$ cm$^{-3}$, $k_B T_e \simeq k_B T_i \simeq 10$ eV, $k_B T_d \simeq 1$ eV, and $Z_d \simeq 10^4$ [24,25]. For simplicity, we have considered for our numerical analysis $\alpha = 1.0$, $\delta = 0.0003$, $\mu_i = 1.4$, $\mu_p = 0.3$, and $\sigma = 1$.

Now, the modulationally stable and unstable parametric regimes of the DAWs can be seen from Figure 1. It is clear from the left panel of Figure 1 that (a) the DAWs are stable for small values of $k$ ($k < k_c$) and are unstable for large values of $k$ ($k > k_c$) under consideration of a sub-extensive $q$; (b) when $q = 1.1$, 1.5, and 1.9, then the corresponding $k_c$ value is $k_c \equiv 3.0$ (dotted blue curve), $k_c \equiv 3.2$ (dashed green curve), and $k_c \equiv 3.4$ (solid red curve); (c) the $k_c$ increases as $q$ increases. Similarly, the right panel of Figure 1 represents the variation of the $P/Q$ with $k$ for different values of $q$ under consideration of a super-extensive $q$: (a) when $q = -0.9$, $-0.5$, and $-0.1$, then the corresponding $k_c$ value is $k_c \equiv 1.40$ (dotted blue curve), $k_c \equiv 1.55$ (dashed green curve), and $k_c \equiv 1.95$ (solid red curve); (b) a negative $q$ also leads to an increase in the critical wave number. Thus, one can say that the direction of the variation of the critical wave number is independent (dependent) of the sign (magnitude) of the $q$.

The modulationally unstable parametric regime (i.e., $P/Q > 0$) of the DAWs allows the generation of highly energetic DA-RWs [54,55]:

$$\phi(\xi, \tau) = \sqrt{\frac{2P}{Q}}\left[\frac{4(1 + 4iP\tau)}{1 + 16P^2\tau^2 + 4\xi^2} - 1\right]\exp(2iP\tau). \tag{23}$$

We have also numerically analyzed Equation (23) in Figure 2 to illustrate the influence of the number density and charge state of the plasma species on the formation of DA-RWs associated with DAWs in the unstable parametric regimes (i.e., $P/Q > 0$). It is clear from the left panel of Figure 2 that (a) the amplitude and width of the DA-RWs decreases with increasing positron number density ($n_{p0}$) for a constant value of $Z_d$ and $n_{d0}$; (b) the increasing negative dust number density ($n_{d0}$) enhances the amplitude and width of the DA-RWs for a constant value of $n_{p0}$ and dust charge state $Z_d$. The right panel of Figure 2 describes the variation of amplitude and width of the DA-RWs with space for different values of $\mu_i$, and it is obvious from this figure that (a) the height and width of the DA-RWs increase (decrease) with an increasing dust (ion) number density for a fixed value of $Z_d$ and $Z_i$; (b) similarly, the nonlinearity of the EPIDPM increases (decreases) with the charge state of the negative dust grains (positive ion) when their number densities remain constant.

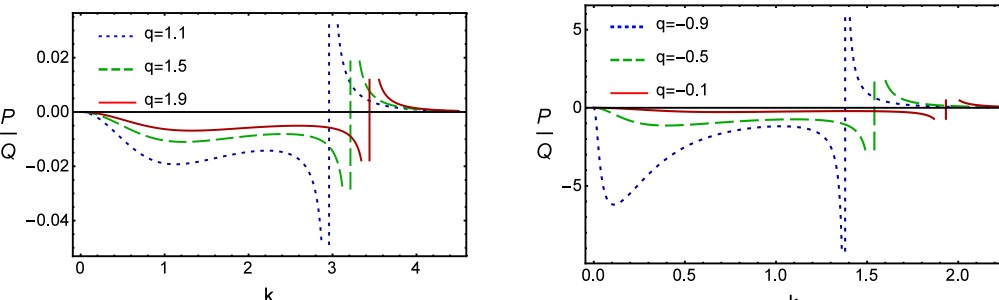

**Figure 1.** Plot of $P/Q$ versus $k$ for different values of positive $q$ (**left** panel), and negative $q$ (**right** panel) when $\alpha = 1.0$, $\delta = 0.0003$, $\mu_i = 1.4$, $\mu_p = 0.3$, and $\sigma = 1$.

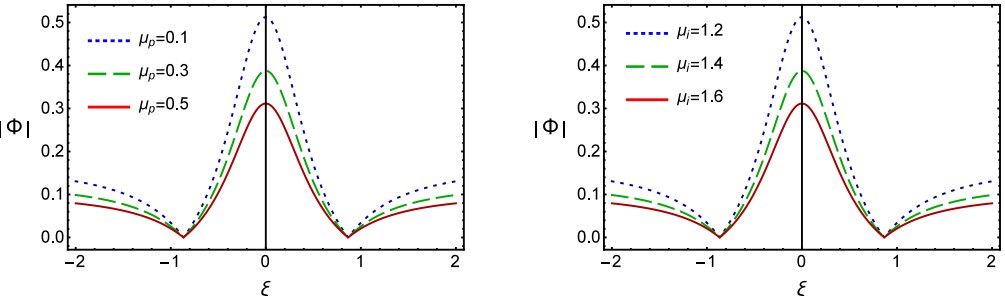

**Figure 2.** Plot of $|\phi|$ versus $\xi$ for different values of $\mu_p$ and $\mu_i = 1.4$ (**left** panel), and plot of $|\phi|$ versus $\xi$ for different values of $\mu_i$ and $\mu_p = 0.3$ (**right** panel) when $\alpha = 1.0$, $\delta = 0.0003$, $q = 1.5$, and $\sigma = 1$.

## 4. Conclusions

In this paper, we have investigated DW-RW-associated DAWs in an EPIDPM with inertial negative dust grains and inertialess non-extensive electrons, positrons, and ions. The dynamics of the EPIDPM and the DA-RWs are governed by the standard NLSE. The results that have been found in our present investigation can be summarized as follows:

- Both modulationally stable (i.e., $P/Q < 0$) and unstable (i.e., $P/Q > 0$) DAWs are observed;
- The direction of the variation of the critical wave number is independent (dependent) of the sign (magnitude) of the $q$;
- The amplitude of the DA-RWs decreases with increasing population of non-extensive positrons;
- Excess non-extensive ions reduce the height of the DA-RWs.

The results of our present investigation will be useful in future to understand the MIs of DAWs and associated DA-RWs in the galactic centre [1], Saturn's magnetosphere [2,3], Jupiter's magnetosphere [3,4], the pulsar magneto-sphere [5–8], supernova environments [8,9], interstellar medium [10–14], cometary tails [15–17], the solar atmosphere [17–20], and laboratory experiments [21–25].

**Author Contributions:** All authors contributed equally to complete this work. All authors have read and agreed to the published version of the manuscript.

**Funding:** The research received no external funding.

**Institutional Review Board Statement:** Not applicable.

**Informed Consent Statement:** Not applicable.

**Data Availability Statement:** Data sharing not applicable—no new data generated.

**Acknowledgments:** The authors are grateful to anonymous reviewers for their constructive suggestions which have significantly improved the quality of our manuscript.

**Conflicts of Interest:** The authors declare no conflict of interest.

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
