# Peer review of "Dust-Acoustic Rogue Waves in an Electron-Positron-Ion-Dust Plasma Medium"

_galaxies, doi:10.3390/galaxies9020031_

Round 1

Reviewer 1 Report

In order to make the work more understandable, it is necessary to take into account the following notes.

The initial physical equations of the model before switching to dimensionless variables should be presented in the paper. The parameters of the object under study (the number density of positrons, electrons, ions and charged dust particles, the charges of ions and dust particles, their masses, temperatures, etc.) should be indicated. It is the physical quantities that are given in the figures, but not the dimensionless ones.

The authors should explain what forms the Tsallis distribution for electrons and why positrons have a different, equilibrium energy distribution. The authors should make estimations of the characteristic times of electron thermalization in such a system, the characteristic times of charging dust particles, and the characteristic times of energy exchange among various components of the plasma.

In a plasma in which positron generation is possible, there must be a powerful field of photons. How does this field of photons affect the very existence of dust particles?

Author Response

We have included a pdf in which authors' response are presented.

Reviewer 2 Report

In the present paper, authors made a theoretical investigation of dust acoustic waves in electron-positron-ion dust medium. The work is motivated by the importance of their results for astrophysical plasmas. The paper is well written, the results represent an advance in the field and merit publication in Galaxies. I have one comment to make the paper little more accessible to a general Galaxies readers :

To show the importance of their results for astrophysical plasmas investigation, the authors refer to [1] and [2] in which other references are given. It would be better that authors refer directly to the orginal references.

Author Response

(The authors gave the same response as above.)

Reviewer 3 Report

The authors present a theoretical investigation about the dust-acoustic (rouge) waves. On the basis of the nonlinear Schrödinger equation, the authors have determined the stable and unstable parametric regions of dust-acoustic (rouge) waves.

I have no comments regarding the theoretical derivations. In the conclusion, the authors state that the obtained results might be useful to understand the nonlinear phenomena in various space dusty plasmas (Lines 196-202). I recommend the authors give more discussions about the applications of the obtained results in those different fields. At least, the readers should have interest on the typical values of those key theoretical parameters in the different fields.

Author Response

(The authors gave the same response as above.)

Round 2

Reviewer 1 Report

The authors tried to take into account all the comments made and did so with varying success. I hope that someone will find the article interesting.